# Influence of Perinatal Factors on Blood Tryptase and Fecal Calprotectin Levels in Newborns

**DOI:** 10.3390/children10020345

**Published:** 2023-02-10

**Authors:** Justine Paysal, Charlotte Oris, Ugo Troin, Pierre-Nicolas Limeri, Jeanne Allard, Marie Tadrent, Bruno Pereira, Etienne Merlin, Emmanuelle Rochette, Bertrand Evrard, Julie Durif, Vincent Sapin, Maguelonne Pons

**Affiliations:** 1CHU Clermont-Ferrand, Néonatologie et Réanimation Pédiatrique, F-63000 Clermont-Ferrand, France; 2INSERM, CIC 1405, CRECHE Unit, CHU Clermont-Ferrand, F-63000 Clermont-Ferrand, France; 3CHU Clermont-Ferrand Gabriel Montpied, Biochimie, F-63000 Clermont-Ferrand, France; 4CHU Clermont-Ferrand, Direction de la Recherche Clinique et de l’Innovation, Secteur Biométrie et Médico-économi, F-63000 Clermont-Ferrand, France; 5CHU Clermont-Ferrand, Pédiatrie, F-63000 Clermont-Ferrand, France; 6CHU Clermont-Ferrand, Immunologie, F-63000 Clermont-Ferrand, France; 7CHU Clermont-Ferrand, Chirurgie Pédiatrique, F-63000 Clermont-Ferrand, France

**Keywords:** blood tryptase, fecal calprotectin, newborns, perinatal factors

## Abstract

Background: Blood tryptase and fecal calprotectin levels may serve as biomarkers of necrotizing enterocolitis. However, their interpretation may be hindered by the little-known effects of perinatal factors. The aim of this study was to compare the tryptase and calprotectin levels in newborns according to their term, trophicity, and sex. Method: One hundred and fifty-seven premature newborns and 157 full-term newborns were included. Blood tryptase and fecal calprotectin were assayed. Results: Blood tryptase levels were higher in premature than in full-term newborns (6.4 vs. 5.2 µg/L; *p* < 0.001). In situations of antenatal use of corticosteroids (*p* = 0.007) and non-exclusive use of human milk (*p* = 0.02), these levels were also higher. However, in multiple linear regression analyses, only prematurity significantly influenced tryptase levels. Fecal calprotectin levels were extremely wide-ranging and were much higher in female than in male newborns (300.5 vs. 110.5 µg/g; *p* < 0.001). Conclusions: The differences in tryptase levels according to term could be linked to early aggression of the still-immature digestive wall in premature newborns, in particular, by enteral feeding started early. The unexpected influence of sex on fecal calprotectin levels remains unexplained.

## 1. Introduction

Necrotizing enterocolitis (NEC) is one of the most frequent and fatal gastrointestinal (GI) disorders in neonatal intensive care [1,2]. Its incidence is inversely proportional to gestational age (GA), with 90% of cases occurring in premature newborns and up to 90% in infants with low birth weight (BW) [2]. Several risk factors have been reported, including GA and BW, as well as intestinal dysbiosis and formula feeding. NEC is considered an inflammatory bowel disease (IBD) characterized by an immature innate immune response [1,2,3,4,5]. However, despite better knowledge of NEC etiology, preventive and therapeutic strategies remain limited [2,6]. Therefore, new tools are required to facilitate an early, precise diagnosis and optimize the management of preterm neonates with intestinal symptoms [6]. However, there is currently a paucity of reliable and robust biomarkers for neonatal enteropathies [4,6].

The tissue residential immune cells, such as mast cells (MCs) and macrophages, together with infiltrated immunocytes, e.g., lymphocytes, dendritic cells, and neutrophils, around the epithelium, endothelium, and intestinal lamina propria, co-work on intestinal immunoregulation and homeostasis of the mucosal immune barrier [1]. MCs, in particular, play an important role [7]. A broad spectrum of inflammatory mediators, including chymase, tryptase, and histamine, are stored in the cytoplasmic granules of MCs and released during MC activation/degranulation by various stimuli [1]. Excessive or chronic MC activation is detrimental and is implicated in several prevalent human GI disorders, including food allergies and IBD [7,8]. Histological studies of NEC specimens from autopsies and surgeries have found that the frequency of submucosal MCs is higher than in normal specimens, often with MC degranulation [9]. Thus, the release of cytokines and chemokines by MCs may also play a significant role in NEC [9]. For example, tryptase, which can upregulate adhesion molecules, increase vascular permeability, recruit eosinophils, and increase epithelial and fibroblast proliferation [7,10], could be involved in the onset of NEC.

Calprotectin, a dimer composed of the protein monomers S100A9 and S100A8, which forms in the cytosol of neutrophils and accounts for up to 60% of the soluble proteins [11], may also be involved in NEC. In the neonatal period, calprotectin regulates the development of the intestinal microbiota and immune system [11,12]. Fecal calprotectin is used later in children and adults as a marker for IBD [4], with a body of evidence correlating calprotectin with IBD, cow-milk allergy, atopic disease, and other GI disorders [11]. A strong correlation has been demonstrated between fecal calprotectin and indium-111-labeled granulocyte scintigraphy (gold standard method for detecting inflammatory activity in IBD) [13]. Nevertheless, its role as a marker of inflammation remains to be confirmed in newborns [11]. Tryptase and calprotectin could thus be implicated in NEC and be used as biomarkers of this disorder. However, interpretation could be difficult owing to the suspected effects of gestational age (GW) on tryptase [9] and calprotectin [14,15,16,17] levels. Moreover, birth weight (BW) and sex could impact these levels [18,19] but their potential effects have never been demonstrated in a neonatal population. Perinatal characteristics related to the course of pregnancy and childbirth, as well as the feeding type, may also influence levels of tryptase and calprotectin but this remains to be explored.

In this context, our study aimed to compare the levels of blood tryptase and fecal calprotectin of newborns according to their birth term (BT), trophicity, and sex and then assess the potential impact of perinatal factors on these levels. Calprotectin is involved in the differentiation [13] and determination of the phenotype of macrophages [12], the producers of tryptase. It was, therefore, relevant to look for a correlation between tryptase and calprotectin levels. We hypothesized that (i) tryptase and calprotectin levels would be influenced by BT and trophicity and that there would be a correlation between these levels and BT, (ii) similar to children and adults [18,19], tryptase levels would be higher in females but calprotectin levels would be independent of sex, (iii) other perinatal characteristics would influence tryptase and calprotectin levels, and (iv) tryptase and calprotectin levels would be correlated.

## 2. Materials and Methods

### 2.1. Study Population

Premature and full-term (FT) newborns recorded between January 2021 and January 2022 were included in this prospective study. Premature newborns were consecutively included for 8 months. Out of 210 premature newborns, 11 did not meet the inclusion criteria (congenital malformations, chromosomic anomalies, and digestive pathologies other than NEC), there were 13 cases of non-consent, and 29 samples were unavailable (insufficient amounts, technical problems). Full-term (FT) newborns were included consecutively for one month. Among the 280 newborns, 9 did not meet the inclusion criteria (congenital malformations, chromosomic anomalies, and digestive pathologies), there were 36 cases of non-consent, and 78 samples were unavailable. Newborns were considered premature if born before 37 gestational weeks (GW) and were classified into three groups: less than 28 GW (Group A), between 28 GW and 31 GW and 6 days (Group B), and between 32 GW and 36 GW and 6 days (Group C). Gestational weeks were calculated as amenorrhea weeks. Premature newborns under 22 GW and weighing less than 500 g were excluded. Full-term newborns (Group D) were recruited in two stages: a first cohort for tryptase assays and a second for calprotectin (tryptase and calprotectin populations). Newborns were considered hypotrophic when their BW was below the 10th percentile.

### 2.2. Ethics

All parents gave written informed consent for their participation. The study was conducted in accordance with the Declaration of Helsinki, and the research protocol for this study was previously approved by our institutional review board the Ethics Committee of CPP-Sud-Est VI (protocol code 2021/CE 26 and date of approval was 4 May 2021).

### 2.3. Perinatal Characteristics

The characteristics of the pregnancy (presence of maternal vascular pathology or diabetes, maternal smoking, antenatal use of corticosteroid, multiple pregnancies), birth (mode of delivery, risk of infection, maternal antibiotic therapy, adaptation to extrauterine life represented by Apgar score, cord pH, and lactates), and type of feeding (exclusive human milk or not) were collected.

### 2.4. Biological Data

Tryptase was assayed from a blood sample placed into EDTA K2 Vacutainer^®^ tubes (Becton Dickinson, Franklin Lakes, NJ, USA) during a systematic check-up (at 12 h of life (H12) for premature newborns and on Day 3 (D3) for the Guthrie test for full-term newborns. It was assayed using a fluoroenzymatic technique (immunoCAP method) at the biochemistry laboratory of Clermont-Ferrand University Hospital.

The fecal calprotectin assay was also performed by the biochemistry laboratory on a meconium sample weighing at least 1 g by immunofluorescence using a Liaison XL Dia analyzer (Diasorin, Italy).

### 2.5. Statistical Analysis

Continuous data were expressed as a median and interquartile range (IQR) according to their statistical distribution. The assumption of normality was assessed using the Shapiro–Wilk test. Comparisons of continuous data (particularly for tryptase and calprotectin) between groups were performed using the nonparametric Kruskal–Wallis test, as the assumptions to apply a parametric test were not met. All perinatal characteristics that significantly impacted tryptase and fecal calprotectin levels were subsequently analyzed together in multivariate linear regression analyses. A logarithmic transformation was applied to assess a normality distribution of tryptase and calprotectin values. A correlation was determined between blood tryptase and fecal calprotectin by linear regression analysis and the Pearson correlation coefficient. *R*^2^ was calculated.

The sample size was estimated in order to assess whether tryptase and calprotectin levels were influenced by BT: premature vs. full-term newborns. With at least 100 infants per group, a minimal difference effect size of 0.5 could be highlighted between groups with a two-sided type I error rate of 5% and a statistical power greater than 90%.

Statistical analyses were performed using Medcalc (version 19.1, Medcalc Software) and GraphPad prism (version 8.0.1). All statistical tests were conducted for a two-sided type I error rate of 0.05.

## 3. Results

### 3.1. Population Characteristics

One hundred and fifty-seven premature newborns (20 in Group A, 43 in Group B, and 94 in Group C) and 157 full-term newborns (57 in the tryptase population and 100 in the calprotectin population) were included. BT, BW, sex, and perinatal characteristics were collected for each newborn, as shown in Table 1.

As expected, there were more multiple pregnancies, maternal vascular pathologies (*p* < 0.001), maternal diabetes (*p* = 0.001), and hypotrophy (*p* = 0.018) in the preterm group than in the full-term group.

Premature newborns were more often born by cesarean section (*p* < 0.001) and had a higher risk of infection (*p* < 0.001) so per partum antibiotic therapy was more frequently used (*p* < 0.001). Antenatal corticosteroids were widely used in premature newborns but no full-term newborns received them. The Apgar score was lower in preterm newborns (Groups A, B, and C) than in full-term newborns (*p* < 0.001). The veinous pH was higher in Group A than in Groups B, C, and D (*p* = 0.01). The arterial pH and lactate levels were not statistically different among the groups.

The use of human milk was more frequent in premature newborns than in full-term newborns (*p* < 0.001). As recommended, the premature newborns in Group A were all fed human milk.

### 3.2. Tryptase

The tryptase levels in our population according to term, trophicity, and sex are presented in Figure 1A. Tryptase levels were higher in premature newborns than in full-term newborns (*p* < 0.001), whereas no differences were observed between hypotrophic and eutrophic or male and female newborns.

Tryptase levels by term (Groups A, B, C, and D) are shown in Figure 1B. In the group of premature newborns (Groups A, B, and C), the tryptase level tended to increase with the advancing term. Group C, therefore, had a higher tryptase level than Groups B and A but the differences were not significant. Unexpectedly, full-term newborns had significantly lower levels than premature newborns in Groups B (*p* = 0.04) and C (*p* < 0.001). There was no significant difference between Groups A and D.

The effect of each of the perinatal characteristics on the tryptase level was also assessed. A significant increase in tryptase levels was found in situations of antenatal use of corticosteroids (*p* = 0.007) and non-exclusive use of human milk (*p* = 0.03) (Figure 2). In multiple linear regression analyses, taking into account the presence or not of prematurity, BT, antenatal use of corticosteroids, and type of enteral feeding, only the presence of prematurity significantly influenced the tryptase level (*p* = 0.019). In this model, the use of antenatal corticosteroids (*p* = 0.379) and type of feeding (*p* = 0.370) did not affect the tryptase level.

### 3.3. Calprotectin

Fecal calprotectin levels in our population according to term, trophicity, and sex are presented in Figure 3. These levels were found to be extremely wide-ranging, with extreme values of 5 and 4360 µg/g. Calprotectin levels were higher in female than in male newborns (*p* < 0.001). This difference was also found in the subgroup analysis in full-term, premature, eutrophic, and hypotrophic newborns studied separately (*p* < 0.001, results not shown). There was no difference in calprotectin levels between premature and full-term newborns or between hypotrophic and eutrophic newborns. Indeed, for blood tryptase, levels were higher in premature newborns than in full-term newborns (6.4 vs. 5.2 µg/L; *p* < 0.001) with an effect size of 0.58 [0.27; 0.90]. For calprotectin, no statistical difference was shown, with a non-clinically relevant effect size of 0.13 [−0.39; 0.13].

The effect of each of the perinatal characteristics on the fecal calprotectin level was assessed independently. None of the perinatal characteristics affected calprotectin levels.

No significant correlation was observed between blood tryptase and fecal calprotectin levels in premature newborns (Figure 4).

## 4. Discussion

The aim of our study was to compare the levels of blood tryptase and fecal calprotectin in newborns according to their BT, trophicity, and sex. We observed that (i) tryptase levels were higher in premature newborns born between 28 GW and 36 GW and 6 days than in full-term newborns, but this difference was not found for very premature newborns under 28 GW. Therefore, no correlation was found between tryptase level and BT; (ii) trophicity and sex did not influence the level of tryptase; and (iii) fecal calprotectin levels were highly variable and higher in female than in male newborns. No other variable impacted this level, and, in particular, there was no effect of the BT, contrary to our initial hypothesis.

### 4.1. Tryptase

The first salient result of this study is that premature newborns had higher tryptase levels than full-term newborns. This difference could be linked to early aggression of the still-immature digestive wall in premature newborns, for example, by enteral feeding started early or due to poor blood perfusion, secondary to persistent ductus arteriosus. This aggression, representing a stress for the organism, could thus activate MCs [7] involved in the host’s innate immune system and trigger the release of protease, including tryptase [1,8,20,21]. Belhocine W. et al. reported a significantly higher serum tryptase level in children aged 0 to 3 months compared to children aged 6 to 12 months, with a median of 6.17 µg/L before 1 month and 3.15 µg /L between 11 and 12 months of life [22]. The decrease is progressive until the age of 12 months and then the rate remains stable until 14 years. This high rate in young children remains to be interpreted according to clinical data, but the hypothesis of increased mast-cell activity in very young children may be an important element in understanding early immunity. Thus, mast cells could be more importantly involved in the immunity of premature babies and their preponderance at the bronchial and digestive levels could be taken into account in the frequent pathologies of this population, namely bronchial dysplasia and enterocolitis. However, at variance with our hypothesis, tryptase levels were not correlated with BT, with premature newborns below 28 GW having levels similar to those of full-term newborns. This raises the question of the degree of maturity of the immune system in premature newborns. Interestingly, Peterslund et al. have demonstrated the presence of MCs from 12 GW and 4 days and their ability to degranulate from 24 GW in NEC [9,23]. The lower performance of MCs in premature newborns below 28 GW could explain our observations [24,25].

These results are also of particular interest because, for the first time, tryptase reference values are proposed for newborns. A previous study analyzed tryptase in the pediatric population but only from age 10 days [26].

In our study, we also observed that tryptase levels were significantly increased in situations of antenatal corticosteroid use, which are closely linked to the occurrence of prematurity. Interestingly, in the multivariate analysis, this significant effect of antenatal corticosteroids was no longer found, demonstrating that antenatal corticosteroids do not directly impact the tryptase level but indirectly through prematurity.

Our second main finding was that exclusive feeding with human milk was associated with lower tryptase levels. This is in line with the well-known immunomodulatory effects of breast milk [2,24,27,28], which, by attenuating inflammatory responses, could limit MC degranulation and tryptase release. Moreover, the aggression of the digestive wall is probably less marked with human milk. In this sense, preterm infants who are not fed breast milk are more susceptible to NEC [2,28]. However, again in multivariate analysis, this result was no longer obtained, with a weak and non-significant effect of the type of milk used. Nevertheless, this observation is possibly linked to the insufficient number of subjects included in this study.

Another important finding is that tryptase levels were not affected by trophicity or sex. To the best of our knowledge, only one study has previously shown that tryptase levels are increased in females, but in adults and not in newborns [19]. Our study is the first to study these levels in a neonatal population.

Although previous work has demonstrated that relative hypoxia stimulates MC degranulation and, therefore, increases tryptase levels [29], our analyses did not reveal any increase in tryptase levels in situations with neonatal hypoxia (maternal vascular pathology, neonatal maladaptation reflected by Apgar, cord pH, or lactates). This difference could be explained by a duration of hypoxia that was not long enough to cause MC degranulation. Additionally, the hypoxia patterns previously assessed were highly specific (retinopathy of prematurity) [29]. A study on newborns treated with hypothermia for anoxo-ischemia could be of interest in this context.

Finally, it is important to note that to avoid adding an unplanned blood sample, we assayed tryptase at H12 for premature births and on D3 at the same time as the Guthrie test for full-term newborns. This leads us to question the impact of this difference in timing on our results. Future studies on the kinetics of blood tryptase in the first days of life could be interesting.

### 4.2. Calprotectin

The activation of primary immunity cells (monocytes, macrophages, neutrophils) located in the intestinal wall releases inflammatory proteins including calprotectin into the digestive lumen. Thus, calprotectin in the stool or fecal calprotectin can be measured [30]. As this protein is remarkably resistant to proteolysis and colonic bacterial degradation, it can be easily quantified in feces [13,31]. However, in our population, as in others [6,32,33,34,35], we observed extremely wide-ranging levels, hindering our interpretation. This raises the question of which perinatal factors might influence calprotectin levels and thus explain this significant inter-individual variability.

First, and unlike other authors [17,36], we observed that fecal calprotectin levels were significantly influenced by sex, with higher levels in females. To the best of our knowledge, only one study has also found a significant increase in urinary calprotectin levels in female compared to male children, with median values of 1351 ng/mL and 31 ng/mL, respectively [37]. This major difference, which was found in urine and children older than those in our study (2–15 years), was unfortunately not explained [37] and no studies have reported on the influence of sex hormones on calprotectin. We note that, interestingly, it has been shown that the composition of meconium varies according to a newborn’s sex [38]. However, the reasons for our observation and its significance remain to be elucidated.

At variance with our hypothesis, we did not find any effect of BT, a potential effect that is highly controversial in the literature. Some authors have reported an increase in fecal calprotectin in premature newborns [15,36] and an impact of the GA of premature births on its levels [14,17], whereas others have not [31,32,39]. However, none of these studies found a clear pattern between GA and calprotectin levels.

The effect of BW is also controversial. Some studies, but not all [31], have demonstrated a negative correlation between BW and fecal calprotectin [32,36]. In our study, we found no difference according to BW or trophicity, which is consistent with the observation that fetal growth does not impact calprotectin [36].

Finally, other perinatal characteristics could explain this marked variability: the mode of delivery (higher levels if vaginal) [12,40,41], causes of prematurity (higher levels if maternal causes) [34], neonatal adaptation (negative correlation with Apgar score) [36,42], type of milk used (higher levels if breastfeeding) [40,41,43,44], and postnatal age [14,40] have all been shown to affect calprotectin. Conversely, some studies [32,36], including ours, do not report these effects. These differences could be explained by the fact that we used a technique automated by Liaison^®^ XL, DiaSorin (calprotectin concentration was determined using a sandwich-type immunoassay), rather than the ELISA technique. Moreover, in our study, calprotectin was measured only in the first stool of a newborn, i.e., meconium. This tells us about prenatal development and exposures but nothing or very little about the postnatal environment (mode of delivery, neonatal adaptation, type of milk used) [38]. However, neither did we observe any difference in calprotectin levels according to antenatal factors such as smoking, maternal vascular pathologies, or as in other studies, [36,41] gestational diabetes, antenatal corticosteroids or antibiotics use. The high levels of calprotectin in the first stool, with no effect of maternal antenatal factors or diet, support the hypothesis that the digestive secretion of calprotectin is a reaction phenomenon to the bacterial digestive colonization of the newborn. An inverse correlation exists between the level of fecal calprotectin and the diversity of the microbiota [11], demonstrating that in neonates, calprotectin regulates the development of the intestinal microbiota and immune system [11,12]. Interestingly, several studies including Warner’s have shown that an imbalance of the intestinal microbiota precedes the occurrence of NEC [45]. The importance of calprotectin for the early development of harmonious microbiota is not fully established but could be one of the explanations for its role in the pathophysiology of NEC. Future studies are needed to better understand the link between calprotectin and the gut microbiota in the context of NEC. Generally, routine use of fecal calprotectin levels seems difficult in the neonatal population, due to the high variability of these values, as demonstrated. In this context, an evaluation of the kinetics of calprotectin based on personalized monitoring would be relevant. Unusual kinetics of the calprotectin level could indicate the occurrence of NEC better than an absolute value alone, which is difficult to interpret. A study has explored this topic, suggesting that there is an inverse relationship between the level of calprotectin and the number of days of life but this remains to be confirmed (low number of subjects were followed) [17].

Calprotectin is involved in the differentiation [13] and determination of the phenotypes of macrophages [12]. Furthermore, the loss of calprotectin in mice altered the phenotypes of colonic lamina propria macrophages compared to wild-type mice [12]. Consequently, as macrophages are the main producers of tryptase, we looked for a correlation between tryptase and calprotectin levels. However, this assessment was not possible in full-term newborns because we had two different cohorts: one for tryptase and another for calprotectin. In premature newborns, we did not find any correlations, probably because we specifically studied fecal and not blood calprotectin. In addition, calprotectin can cause macrophage differentiation but not macrophage degranulation. Therefore, a correlation may be present between the levels of calprotectin and macrophages but not between calprotectin and tryptase, which only reflects the number of degranulated macrophages.

A limitation of this study is that it is descriptive only and does not explore associations between these biomarkers and NEC and perinatal characteristics. Larger studies are, therefore, required to determine more precisely the effects of calprotectin and tryptase on the occurrence of NEC and their possible use as biomarkers. Moreover, an evaluation of the kinetics of these two proteins would be interesting in terms of assessing the impact of the perinatal factors explored in this study as a function of time, at a distance from birth, as well as assessing these kinetics for the prediction of NEC.

## 5. Conclusions

Our study shows that blood tryptase is influenced by BT, with premature newborns (between 28 and 36 GW and 6 days) having higher levels than full-term newborns. Conversely, fecal calprotectin does not appear to be affected by BT but rather unexpectedly by sex. Further studies would help to better understand the mechanisms underlying our results. Moreover, our study is the first to provide reference values for these assays. It would, therefore, be worthwhile to evaluate the utility of these assay values as NEC biomarkers in premature newborns.

## Figures and Tables

**Figure 1 children-10-00345-f001:**
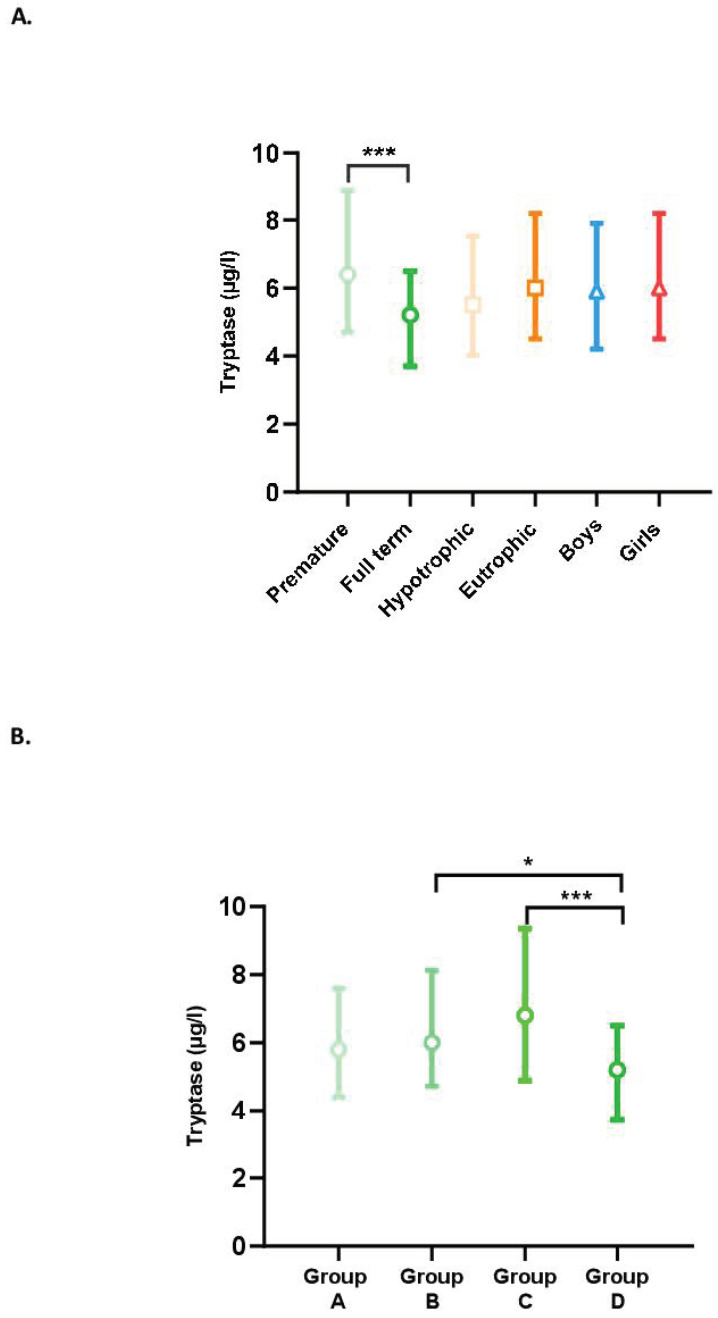
(**A**) Blood tryptase levels according to the characteristics of the newborns (term, trophicity, sex). Difference between groups: *** *p* < 0.001. (**B**) Impact of term on tryptase levels. Difference between groups: (A) less than 28 GW; (B) between 28 GW and 31 GW and 6 days; (C) between 32 GW and 36 GW and 6 days; and (D) full-term newborns; * *p* < 0.05, *** *p* < 0.001.

**Figure 2 children-10-00345-f002:**
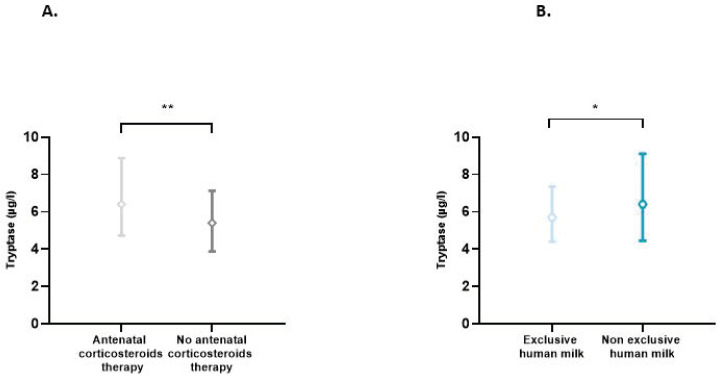
(**A**) Blood tryptase according to the use or not of antenatal corticosteroids therapy. Difference between groups: ** *p* < 0.01. (**B**) Impact of exclusive human milk diet on blood tryptase levels. Difference between groups: * *p* < 0.05.

**Figure 3 children-10-00345-f003:**
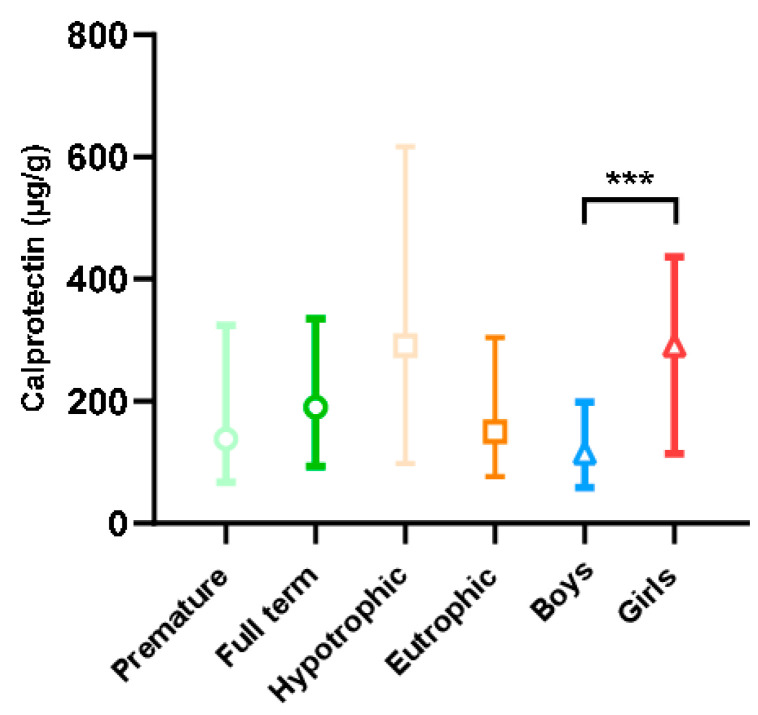
Fecal calprotectin level according to the characteristics of the newborns (term, trophicity, sex). Difference between groups: *** *p* < 0.001.

**Figure 4 children-10-00345-f004:**
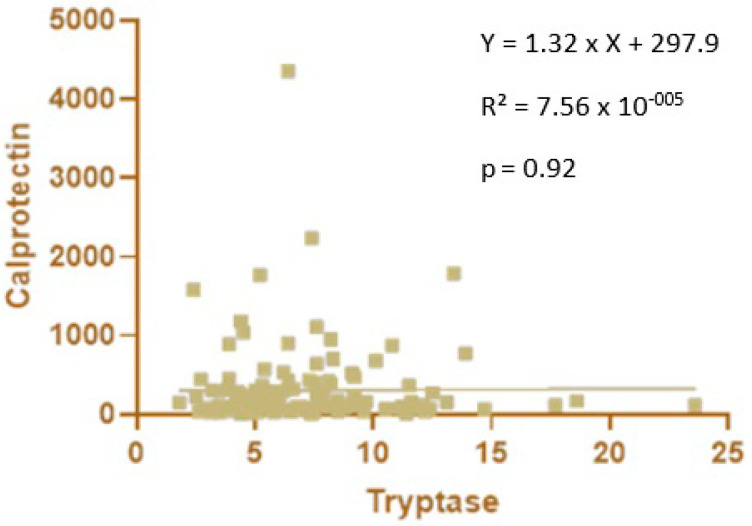
Correlation between blood tryptase and fecal calprotectin levels in premature newborns.

**Table 1 children-10-00345-t001:** Patient characteristics. *: significantly different (*: *p* < 0.05; **: *p* < 0.01; ***: *p* < 0.001).

	Premature Newborns (*n* = 157)			Full-Term Newborns: Group D (*n* = 157)	
	Group A <28 GW (*n* = 20)	Group B 28–<32 GW (*n* = 43)	Group C 32–<37 GW (*n* = 94)	Calprotectin Population (*n* = 100)	Tryptase Population (*n* = 57)
**Multiple pregnancies (%) *****	15	12	13	0	0
(twin/triplet)					
**Maternal vascular pathology (%) *****	30	30	23	1	10
**Maternal diabetes (%) ****	15	26	23	6	21
**Maternal smoking (%)**	10	19	15	17	21
**Antenatal corticosteroid therapy (%) *****					
Complete	90	67	58	0	0
Incomplete	5	14	13	0	0
Absent	5	19	29	100	100
**Infectious risk *****	35	72	61	29	42
**Maternal antibiotic therapy (%) *****	100	81	93	48	55
**Mode of delivery (%) *****					
Vaginal	30	37	47	86	79
Cesarean	70	63	53	14	21
**Sex**					
Male (%)	50	47	60	55	47
**Birth weight (g) *****	855 (715–945)	1 355 (1190–1555)	2 030 (1785–2398)	3283 (3085–3608)	3 425 (2988–3721)
Hypotrophy (%) *	10	16	21	4	16
**Apgar score M5 *****	7.5 (6–8)	7.5 (7–8)	8 (7–9.5)	10 (10–10)	10 (9–10)
**Arterial pH**	7.30 (7.28–7.36)	7.26 (7.22–7.31)	7.26 (7.21–7.31)	7.25 (7.21–7.30)	7.25 (7.20–7.31)
**Venous pH ***	7.38 (7.32–7.40)	7.31 (7.25–7.36)	7.30 (7.26–7.34)	7.33 (7.28–7.35)	7.32 (7.28–7.35)
**Lactates (mmol/L)**	3.00 (2.30–3.75)	3.05 (2.35–5.10)	3.10 (2.30–4.60)	3.20 (2.40–4.20)	3.85 (2.50–5.40)
**Exclusive human milk (%) *****	100	95	24	52	63

## Data Availability

The datasets used and/or analyzed in this study are available from the corresponding author on reasonable request.

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
