# Peer review of "Influence of Perinatal Factors on Blood Tryptase and Fecal Calprotectin Levels in Newborns"

_children, 2023, doi:10.3390/children10020345_

Round 1

Reviewer 1 Report

Can you please clarify further? I'll break down some parts separately as below..  
  1. Were the parents of the infants approached and recruited consecutively over the study period? 
  2.  
  3. were any infants/families approached who refused to be involved? how many?
  4.  
  5. were any infants/families approached who did not fit study criteria
  6.  
  7. in any of the included cases, had the mothers changed their diet prior to enrolment? or was prior dietary exclusions/changes seen as a factor that excluded the infant from the study?
  8.  
  9. Do a flow diagram that gave the numbers of people approached, number excluded, number refused etc. But these details might be easier to give as text. 
  10.  
  11. mention normal level of fecal calprotectin in newborns
  12.  
  13. did the involved infants have any tests prior to enrolment or was their assessment purely based on their presenting symptoms/features?
  14.  
  15.  

Author Response

Comments and Suggestions for Authors

Can you please clarify further? I'll break down some parts separately as below.. 

* Were the parents of the infants approached and recruited consecutively over the study period?

We have added information in the study population section.  

* were any infants/families approached who refused to be involved? how many?

We have added information in the study population section.

* were any infants/families approached who did not fit study criteria

 We have added information in the study population section.

* in any of the included cases, had the mothers changed their diet prior to enrolment? or was prior dietary exclusions/changes seen as a factor that excluded the infant from the study?

 We did not control or modify the diets of the mothers.

* Do a flow diagram that gave the numbers of people approached, number excluded, number refused etc. But these details might be easier to give as text.

 Yes, we now include this information in the text.

* mention normal level of fecal calprotectin in newborns

In a recent pediatric cohort (n = 174 children), decision thresholds were validated for three distinct age groups: 910 µg/g of stool in children aged less than 1 year, 286 µg/g of stool in children aged 1–4 years, and 54 µg/g of stool in those aged 4–12 years [doi.org/10.1080/00365521.2020.1794026.].

From age 4 years, the reference values coincided, as expected, with those established in adults. A threshold of 648 µg/g has also been validated in a large cohort of children aged less than 1 year (n = 239) [doi.org/10.1080/00365521.2020.1794026.]. Other studies conducted in newborns and infants have reported strong inter-individual variability in calprotectin concentrations, with values ranging from 9 to 2880 µg/g of stool depending on the subject [doi.org/10.1080/00365521.2020.1794026;doi.org/10.3390/jcm9124089; doi.org/10.1097/MPG.0b013e3180320643; doi.org/10.3390/jcm8040473]. There is thus no consensus on a reference value for full-term newborns or premature babies.      

* did the involved infants have any tests prior to enrolment or was their assessment purely based on their presenting symptoms/features?

Children were included from birth if no non-inclusion criteria were met (congenital malformation, chromosomic anomaly, digestive pathologies other than NEC).

Reviewer 2 Report

Authors have highlighted the importance of fecal calprotectin levels and blood tryptase levels on the gestational age of newborn infants.

It is better to use gestational age instead of birth term (BT) as it can confuse the readers.Else give a one line definition for birth term

Authors have not mentioned about how did they derive at the sample size of 157 preterm and 157 term infants.

This calculation is required.

Among the limitations, the need for large sample size is indirectly emphasised.

Author Response

* It is better to use gestational age instead of birth term (BT) as it can confuse the readers. Else give a one line definition for birth term

We have added a definition of birth term in the text.

* Authors have not mentioned about how did they derive at the sample size of 157 preterm and 157 term infants. This calculation is required.

We thank the reviewer for the helpful comment. The sample size was estimated in order to aim that tryptase and calprotectin levels were influenced by the gestational age: premature vs. full-term newborns. With at least 100 infants by group, a minimal difference effect-size of 0.5 could be highlighted between groups for a two-sided type I error at 5% and a statistical power greater than 90%. For blood tryptase, levels were higher in premature newborns than full-term newborns (6.4 vs 5.2 µg/l; p<0.001) with an effect-size at 0.58 [0.27; 0.90]. For calprotectin, no statistical difference was shown with a non-clinically relevant effect-size at 0.13 [-0.39; 0.13].

* Among the limitations, the need for large sample size is indirectly emphasised.

We thank the reviewer for the comment. For the primary analysis aiming to compare tryptase and calprotectin levels between premature and full-term newborns infants, sample size seems sufficient as aforementioned. Indeed, effect size observed for tryptase (0.58 [0.27; 0.90]) was greater than that fixed for sample size estimation equals 0.5. For this comparison, statistical power was satisfactory. For calprotectin, effect-size was low and non-relevant (0.13 [-0.39; 0.13]) according to Cohen’s recommendations which define effect size bounds as small (ES: 0.2), medium (ES: 0.5) and large (ES: 0.8, “grossly perceptible and therefore large”).

We agree that larger sample size would be needed for secondary objectives (such as comparisons among term for premature newborns) which are, in this work, exploratory.

Reviewer 3 Report

The study by Paysal etal studies the association between various maternal and infant characteristics on the blood tryptase levels and fecal calprotectin levels. 

The associations that they highlight are either not strong enough ( a not very linear relationship between infant maturity and tryptase levels) or are controversial ( higher trptase levels associated with antenatal corticosteroids that are supposed to protect from NEC) in explaining the associations between these biomarkers and NEC

This is an important area of research and should have involved larger sample sizes and should have been relegated to the preterm infant population.  

Further well designed studies are required to explore these associations

There are a few structuring /grammatical errors and spelling mistakes in the manuscript that will need attention

Author Response

* The associations that they highlight are either not strong enough (a not very linear relationship between infant maturity and tryptase levels) or are controversial (higher tryptase levels associated with antenatal corticosteroids that are supposed to protect from NEC) in explaining the associations between these biomarkers and NEC

Yes, but our objective was to assess the impact of several perinatal factors on tryptase and calprotectin levels and not to assess the values of these markers as predictors of enterocolitis.

* This is an important area of research and should have involved larger sample sizes and should have been relegated to the preterm infant population. 

You are right, we have added this suggestion in the limits section.

* Further well designed studies are required to explore these associations

You are right and we have added this in the limits.

* There are a few structuring /grammatical errors and spelling mistakes in the manuscript that will need attention

The text has been professionally language-edited.

Round 2

Reviewer 2 Report

The manuscript has been modified as suggested

Reviewer 3 Report

I have no more comments to provide